# Development of a Spectrophotometric Assay for the Cysteine Desulfurase from *Staphylococcus aureus*

**DOI:** 10.3390/antibiotics14020129

**Published:** 2025-01-26

**Authors:** Emily Sabo, Connor Nelson, Nupur Tyagi, Veronica Stark, Katelyn Aasman, Christine N. Morrison, Jeffrey M. Boyd, Richard C. Holz

**Affiliations:** 1Department of Chemistry, Colorado School of Mines, Golden, CO 80401, USA; esabo@mines.edu (E.S.); connornelson@mines.edu (C.N.); christine.n.morrison@gmail.com (C.N.M.); 2Department of Biochemistry and Microbiology, Rutgers University, New Brunswick, NJ 08901, USA; nupur.tyagi@rutgers.edu

**Keywords:** *Staphylococcus aureus*, MRSA, Fe-S cluster biosynthesis, cysteine desulfurase, SUF, enzyme kinetics, cycloserine

## Abstract

**Background/Objectives**: Antibiotic-resistant *Staphylococcus aureus* represents a growing threat in the modern world, and new antibiotic targets are needed for its successful treatment. One such potential target is the pyridoxal-5′-phosphate (PLP)-dependent cysteine desulfurase (*Sa*SufS) of the SUF-like iron–sulfur (Fe-S) cluster biogenesis pathway upon which *S. aureus* relies exclusively for Fe-S synthesis. The current methods for measuring the activity of this protein have allowed for its recent characterization, but they are hampered by their use of chemical reagents which require long incubation times and may cause undesired side reactions. This problem highlights a need for the development of a rapid quantitative assay for the characterization of *Sa*SufS in the presence of potential inhibitors. **Methods**: A spectrophotometric assay based on the well-documented absorbance of PLP intermediates at 340 nm was both compared to an established alanine detection assay and used to effectively measure the activity of *Sa*SufS incubated in the absence and presence of the PLP-binding inhibitors, D-cycloserine (DCS) and L-cycloserine (LCS) as proof of concept. Methicillin-resistant *S. aureus* strain LAC was also grown in the presence of these inhibitors. **Results**: The Michaelis–Menten parameters *k_cat_* and *K_m_* of *Sa*SufS were determined using the alanine detection assay and compared to corresponding intermediate-based values obtained spectrophotometrically in the absence and presence of the reducing agent tris(2-carboxyethyl)phosphine (TCEP). These data revealed the formation of both an intermediate that achieves steady-state during continued enzyme turnover and an intermediate that likely accumulates upon the stoppage of the catalytic cycle during the second turnover. The spectrophotometric method was then utilized to determine the half maximal inhibitory concentration (IC_50_) values for DCS and LCS binding to *Sa*SufS, which are 2170 ± 920 and 62 ± 23 μM, respectively. Both inhibitors of *Sa*SufS were also found to inhibit the growth of *S. aureus*. **Conclusions**: Together, this work offers a spectrophotometric method for the analysis of new inhibitors of SufS and lays the groundwork for the future development of novel antibiotics targeting cysteine desulfurases.

## 1. Introduction

Since their discovery in the mid-20th century, antibiotics have seen near ubiquitous employment in the prophylactic and remedial treatment of bacterial infections in humans, pets, and agricultural animals alike [1,2]. This increased use of antibiotics has facilitated the evolution and pervasive horizontal gene transfer of several thousand unique antibiotic resistance genes [1,2,3,4]. In 2019 alone, infections caused by antimicrobial-resistant pathogens resulted in an estimated 4.95 million global deaths [5]. While this number, likely influenced by the COVID-19 pandemic, decreased to 4.71 million deaths in 2021, it is expected to increase to 8.22 million by 2050 [6]. Of these antimicrobial-resistant pathogens, methicillin-resistant *Staphylococcus aureus* (MRSA) remains one of the most serious [7,8,9]. MRSA strains contain either the mobile *mecA* or *mecC* gene, both of which encode a penicillin-binding protein (PBP) that effectively circumvents the effectiveness of penicillin, methicillin, and other widely used β-lactam antibiotics that target peptidoglycan synthesis [7,8]. A mostly nosocomial infection, MRSA alone is estimated to have directly caused 130,000 of the deaths attributed to antibiotic-resistant bacteria in 2021 [6]. Vancomycin, a glycopeptide antibiotic targeting peptidoglycan cell wall synthesis via a distinct mechanism, has long been used as a last-resort mechanism for treating persistent MRSA [8,9,10,11]. However, its increased application has mediated the rapid accumulation of mobile *van* resistance genes in strains of *S. aureus* [8,10,12]. Indeed, infections caused by vancomycin-intermediate and vancomycin-resistant strains of *S. aureus* (VISA and VRSA, respectively), have seen an increased global prevalence in hospitals over the past three decades [8,9,10,11]. To effectively curb the cost of life induced by this ongoing antibiotic resistance crisis, new antibiotics and antibiotic targets are required to treat bacteria that are otherwise resistant to our current methods of treatment.

Iron–sulfur (Fe-S) clusters, a versatile family of ancient cofactors fundamental to essential biological processes in every domain of life, represent a promising target for the development of antibiotics [13,14,15,16]. Six unique pathways have evolved for Fe-S cluster biosynthesis: the Nirogen Fixation pathway (NIF), the Iron–Sulfur Cluster pathway (ISC), the Cytosolic Iron–Sulfur Cluster Assembly pathway (CIA), the Minimal Iron–Sulfur pathway (MIS), the Sulfur Mobilization Factor pathway (SUF), and the chimeric SUF-like pathway [17,18,19,20,21,22,23]. Each pathway comprises, among other essential and auxiliary proteins, a cysteine desulfurase that secures activated sulfur from L-cysteine (Cys) in the form of a persulfide and a scaffold upon which Fe-S clusters are constructed [21,24]. Of the six aforementioned pathways, the genomes of most Gram-positive bacteria, including that of *S. aureus*, only possess an operon corresponding to the SUF-like pathway [18,25,26]. Indeed, upon artificially decreasing SUF transcriptional activity in *S. aureus*, Roberts et al. [26] observed diminished metabolic activity corresponding to Fe-S cluster-dependent enzymes, as well as an increased susceptibility to oxidative stress, indicative of Fe-S cluster deficiency. This apparent dependence makes the SUF-like pathway of *S. aureus* an ideal target for the development of a novel class of antibiotics, as the successful inhibition of just one of the essential enzymes comprising this pathway may terminate the ability of *S. aureus* to synthesize Fe-S clusters that are required for survival.

Of the enzymes comprising the SUF-like pathway of *S. aureus*, the homodimeric cysteine desulfurase, SufS (*Sa*SufS), represents a promising target for novel antibiotics [26,27,28]. Like other cysteine desulfurase enzymes, SufS is a pyridoxal-5′-phosphate (PLP)-dependent aminotransferase that catalyzes the in vivo conversion of Cys to L-alanine (Ala) [21,22,24]. A testament to the viability of targeting PLP-dependent bacterial enzymes, the PLP-binding antibiotic, D-cycloserine (DCS), has been used to treat multidrug-resistant *Mycobacterium tuberculosis* [29]. DCS was recently shown to covalently bind and inhibit the SufS from the protozoan parasite *Plasmodium falciparum* [28]. Furthermore, DCS and its enantiomer, L-cycloserine (LCS), have been shown to irreversibly bind and inhibit the activity of the cysteine desulfurase from *B. subtilis* (*Bs*SufS) [27]. While DCS and LCS have shown promise as potential inhibitors of SufS, their lack of specificity hampers their capacity as iron–sulfur targeting antibiotics [30]. A major challenge in the discovery of novel inhibitors targeting SufS lies in the enzymatic assays used for SufS (i.e., persulfide detection via the FeCl_3_-catalyzed formation of methylene blue [31,32] or alanine (Ala) detection via the formation of a fluorescent Ala-naphthalene-2,3-dicarboxaldehyde (Ala-NDA) adduct [25,33,34]). Both methods require a chemical quenching step, followed by a long incubation period necessary for the complete development of the respective spectrophotometrically active analyte [25,31,32,33,34]. These assays not only necessitate the use of several reactive compounds, each with the potential to engage in undesired side reactions with unreacted inhibitors, but they also impose limits on the conditions in which the assays themselves can be run.

In an effort to identify new small molecule inhibitors of *Sa*SufS that can potentially function as novel antibiotics, a spectrophotometric assay was developed that allows for the direct detection of *Sa*SufS activity based on previously observed intermediates of the SufS-Cys catalytic cycle. The validation of this spectrophotometric assay was achieved by determining the relative kinetic constants for the formation of these intermediates and by comparing them to each other, as well as *k_cat_* and *K_m_* values derived using the conventional Ala-NDA-based assay. With this spectrophotometric assay in hand, the half maximal inhibitory concentrations (IC_50_ values) of *Sa*SufS for LCS and DCS were determined and compared to those previously reported for SufS enzymes from other organisms, further validating this assay. Promisingly, both *Sa*SufS inhibitors were found to inhibit the growth of *S. aureus*. These results lend credence to the practical spectrophotometric assay reported herein as being broadly applicable to all PLP-dependent aminotransferases with which both kinetic and inhibitor binding information may be obtained. With such an assay, novel antibiotics targeting the cysteine desulfurase of *S. aureus* may be more easily and effectively tested.

## 2. Results

### 2.1. Kinetics of SaSufS with Cys as a Substrate

Initially, an Ala-NDA-based assay was used to determine the Michaelis–Menten kinetic constants, *k_cat_* and *K_m_*, of both *Sa*SufS and the corresponding cysteine desulfurase from the Gram-positive organism, *Bacillus subtilis* (*Bs*SufS) [25]. To benchmark the cysteine desulfurase activities of these enzymes alone, assays were performed in the absence of the cysteine transferase protein SufU. PLP occupancy was quantified for *Sa*SufS by chemically liberating PLP from its protein-bound state and measuring its absorbance at 390 nm [33]. Ala formation corresponding to the cysteine desulfurase activity of both enzymes were quantified using a 96-well plate with a maximum of 1.5 mM Cys. Relative to previous work with this assay [25,33,34], the concentration of NDA was increased to 2 mM such that NDA was in excess of the total concentration of primary amine (Ala-produced and Cys-added), and the concentration of SufS was increased to 50 µM. Increased substrate and NDA concentrations were determined to be critical for the maximum velocity (*V*_max_) to be achieved for the *Sa*SufS-catalyzed cysteine desulfurase reaction. The linear initial velocity of the *Sa*SufS cysteine desulfurase reaction in the presence of the reducing agent, TCEP, was observed from 0 to 2 min (Figure 1a,c). Under these reaction conditions, reproducible kinetic parameters were obtained.

The kinetic parameters for *Sa*SufS, with Cys as the substrate in 100 mM MOPS buffer, and pH 8.0 at 20 °C, were obtained by plotting the specific activity (SA) versus the Cys concentration and fitting these data to the Michaelis–Menten equation (Figure 1b,d). For *Sa*SufS, a PLP-corrected *k_cat_* of 4.1 ± 0.5 min^−1^ and a *K_m_* value of 600 ± 170 µM Cys were obtained, resulting in a catalytic efficiency (*k_cat_*/*K_m_*) for Cys of 6800 ± 2700 min^−1^ M^−1^ (Table 1). For *Bs*SufS, the corresponding PLP-corrected *k_cat_* and *K_m_* of 4.9 ± 0.7 min^−1^ and 440 ± 190 μM Cys, respectively, were determined, resulting in a catalytic efficiency (*k_cat_*/*K_m_*) for Cys of 11,100 ± 6400 min^−1^ M^−1^ (Table 1).

### 2.2. Spectrophotometric Determination of Michaelis–Menten Constants

The UV-Vis spectra of *Sa*SufS were recorded between 300 and 460 nm over the course of 5 min at 22 °C (Figure 2). *Sa*SufS without Cys exhibited a λ_max_ at 420 nm, characteristic of the PLP cofactor bound to Lys250 (internal Lys–aldimine) [27,35,36]. Once the Cys substrate was added to *Sa*SufS, the absorbance at 420 nm diminished in intensity, and a new peak arose at 340 nm with an isosbestic point at ~360 nm.

Michaelis–Menten curves corresponding to both a short-term (0 to 40 s) and a long-term (40 to 120 s) rate of change in the absorbance of *Sa*SufS at 340 nm upon the introduction of different concentrations of Cys were generated (Figure 3). From these curves, the relative *k_cat_* values (*k_int_* values) and *K_m_* values were extracted (Table 1). These parameters were obtained at 12 °C, as at this temperature, the formation of both a short-term intermediate reaching steady state and an accumulating long-term (trapped) intermediate upon the second turnover could be spectrophotometrically observed. In the absence of TCEP, the PLP-corrected *k_int_* values were found to be 350 ± 18 min^−1^ and 205 ± 8 min^−1^ from 0 to 40 s and from 40 to 120 s, respectively. In the presence of TCEP, the *k_int_* values were found to be 350 ± 13 min^−1^ and 173 ± 8 min^−1^ for 0 to 40 s and 40 to 120 s, respectively. The *K_m_* values for Cys between 0 and 40 s were found to be 570 ± 100 and 337 ± 52 μM Cys in the absence and presence of 2 mM TCEP, respectively. Between 40 and 120 s, the *K_m_* values for Cys were found to be 1030 ± 120 μM and 616 ± 93 μM Cys in the absence and presence of 2 mM TCEP, respectively.

### 2.3. Absorbance Spectroscopy of SaSufS with LCS and DCS

*Sa*SufS was incubated with 4.9 mM LCS (Figure 4a). Within the first 30 min, the magnitude of the broad peak corresponding to LLP at ~420 nm diminished in intensity and two new peaks emerged at ~405 nm and ~335 nm, respectively. Over the course of the remaining ~24 h, the absorption bands at ~420 and ~405 nm gradually decreased in intensity with a simultaneous shift in λ_max_ to ~385 nm. Concurrently, the absorption band at ~335 nm increased in intensity. As a result, the *Sa*SufS solution incubated in LCS gradually lost its characteristic yellow color.

Similarly, *Sa*SufS was incubated with 4.9 mM DCS, and its electronic absorption spectrum was recorded every hour for ~24 h (Figure 4b). The PLP-associated peak at ~420 nm gradually decreased in intensity, with a new peak emerging at ~380 nm. No other developing absorbance was observed. Furthermore, a clear isosbestic point at ~405 nm appeared, suggesting no additional intermediates. This change in the UV-Vis spectrum of *Sa*SufS was also accompanied by a visual loss of its characteristic yellow color.

### 2.4. Quantitation of the Efficacy of LCS and DCS as Inhibitors of SaSufS

Dose–response curves were generated after a 96 h incubation in DCS and LCS, allowing for the corresponding IC_50_ values to be extracted (Figure 5). In the presence of 10 mM Cys, *Sa*SufS activity was monitored through the observation of the change in absorbance at ~340 nm upon the addition of varying concentrations of each inhibitor. Since both cycloserine inhibitors react slowly with *Sa*SufS, all protein-inhibitor samples were incubated for 96 h at 20 °C before Cys was added. LCS was found to have an IC_50_ value of 62 ± 23 μM, while DCS was found to have an IC_50_ value of 2170 ± 920 μM (Table 2).

As the use of a spectrophotometric assay to gain information regarding the inhibitors of *Sa*SufS is novel, the IC_50_ value obtained for LCS was verified using the Ala-NDA assay [25,33,34]. Since NDA can react with any primary amine, including cycloserine, the concentration of all primary amines in the solution cannot exceed that of NDA; therefore, the data above 150 µM DCS or LCS could not be obtained using this method. The data obtained using the Ala-NDA assay, normalized using the activity of *Sa*SufS incubated in the absence of any inhibitor, are shown in red for LCS (Figure 5a). These data, when fit to a dose–response curve, yielded an IC_50_ value of 33 ± 12 μM LCS (Table 2).

### 2.5. LCS and DCS Antimicrobial Growth Assays

The hypothesis that LCS and DCS decreased *S. aureus* growth through the inhibition of SUF-directed Fe-S cluster synthesis was tested. For these assays, the community-associated *S. aureus* MRSA strain LAC (wild type; WT) was used. Growth was also examined in the Δ*nfu* Δ*sufT* double mutant strain, which is defective in maturating Fe-S proteins [37,38]. It was determined that LCS and DCS inhibit the growth of *S. aureus* (Figure 6). From these data, in vivo IC_50_ values were obtained for both LCS (4.4 ± 0.7 mM) and DCS (120 ± 13 μM).

## 3. Discussion

SufS enzymes catalyze the desulfurization of Cys and the simultaneous formation of a persulfide on their catalytic Cys residue (Cys389 in *Sa*SufS), yielding Ala as a byproduct (Figure 1). This cysteine desulfurase mechanism begins with a lysine (Lys) residue (Lys250 in *Sa*SufS) bound to the PLP cofactor, forming a Lys–aldimine complex. Cys then displaces the Lys residue and binds PLP to form a Cys-aldimine intermediate. Lys has been proposed to deprotonate the enolimine tautomer of the Cys-aldimine intermediate, forming the Cys-quinonoid species [35]. Lys then protonates the resulting Cys-quinonoid to form the Cys-ketimine intermediate. A mutational analysis of SufS enzymes revealed an active site His residue (His147 in *Sa*SufS) that deprotonates the catalytic Cys residue, allowing for the formation of the persulfide on Cys and yielding an Ala-enamine intermediate [35]. It was suggested that Lys returns to cycle through the Ala-ketimine, Ala-quinonoid, and Ala-aldimine intermediates, eventually releasing Ala and returning to the resting state of SufS (Lys–aldimine) [35]. Importantly, before it can undergo an identical second turnover, the persulfide formed on Cys389 in *Sa*SufS must be reduced to its monosulfurated form. In the absence of such a reducing agent, Nakamura et al. [39] suggested that *Bs*SufS becomes trapped at the Cys-aldimine intermediate, as the corresponding persulfurated cysteine is unable to effectively perform the nucleophilic attack necessary for continued catalysis.

### 3.1. Spectrophotometric Determination of SaSufS Kinetic Constants

A kinetic analysis of the entire catalytic mechanism of cysteine desulfurases, like *Sa*SufS, is typically accomplished using either a methylene blue or an Ala-NDA-based assay [40]. However, the need to quench the enzymatic reaction prior to development of the analyte with these methods poses a challenge for kinetic analysis, particularly in the determination of inhibition constants. Previous work directly analyzed the PLP intermediates of SufS spectroscopically [35,36], which provided a basis for the development of the spectrophotometric assay described herein. Before such an assay could be developed for *Sa*SufS, it was initially verified that, upon incubation in cysteine, the UV-Vis spectrum of *Sa*SufS would quantitatively reveal the development of intermediates spectroscopically distinct from the internal Lys–aldimine complex. Indeed, a partial shift in the absorbance of the PLP cofactor from 420 to 340 nm, as well as an increase in the absorbance at 340 nm on the timescale of ~5 min (Figure 1), revealed the formation of one or more intermediates absorbing at 340 nm. These results are consistent with the data reported for other SufS cysteine desulfurases [35,36,39,41]. Stopped-flow, spectrophotometric, and crystallographic experiments performed by Blahut et al. revealed the formation of an intermediate absorbing at 340 nm, which they identified as Cys-aldimine, just 5 s after the addition of Cys to *Escherichia coli* SufS (*Ec*SufS) [35]. In the presence of a reducing agent, the absorbance at 340 nm diminished and the LLP-associated peak at 420 nm returned to its original intensity after 150 min [35]. These data suggest that the associated *Ec*SufS intermediate was in the steady state during the reaction [35]. Based on their results, they predicted that Cys-aldimine likely represents a bottleneck in the cysteine desulfurase catalytic mechanism of *Ec*SufS and that the formation of Cys-ketimine through a Cys-quinonoid intermediate may be rate limiting [35]. Furthermore, upon soaking crystals of *Bs*SufS in 5 mM Cys in the absence of a reducing agent, Nakamura et al. found that, after just 63 s, the catalytic cysteine residue 361 had been persulfurated and an electron density they ascribed to an Ala-aldimine intermediate was present in the active site [39]. Soaking the crystals for 90 s revealed that a trapped intermediate, which they identified as Cys-aldimine, had appreciably accumulated in the already persulfurated active site [39]. Spectroscopy performed up to 90 s after the incubation of *Bs*SufS with cysteine in the absence of a reducing agent revealed the formation of an intermediate absorbing at 338 nm within 60 s, suggesting that both aldimine intermediates absorb at approximately 340 nm [35,36,39].

*Sa*SufS was initially kinetically characterized in 2 mM TCEP to promote the reduction of the persulfide using the established Ala-NDA assay, and its kinetic constants were compared to those of *Bs*SufS under identical conditions. The average PLP occupancy of *Sa*SufS was found to be 66 ± 4% and for *Bs*SufS 49 ± 12%. Our attempts to supplement purified *Sa*SufS or *Bs*SufS with free PLP did not increase the occupancy. Interestingly, Selbach et al. [25] reported an occupancy of 93 ± 60% for *Bs*SufS, indicating a possible higher PLP occupancy for *Bs*SufS. However, given the large error of their reported occupancy [25], it is likely that both SufS enzymes have similar PLP occupancies. As such, all activities were corrected for the PLP occupancy of the enzyme used. The *k_cat_* and *K_m_* of *Sa*SufS (4.1 ± 0.5 min^−1^ and 600 ± 170 µM Cys, respectively) were similar in magnitude to those of *Bs*SufS (4.9 ± 0.7 min^−1^ and 440 ± 190 μM Cys, respectively), suggesting that both cysteine desulfurases catalyze the conversion of Cys to Ala at similar rates and with similar substrate binding affinities. Indeed, even though the catalytic efficiency of *Bs*SufS (11,100 ± 6400 min^−1^ M^−1^) is greater than that of *Sa*SufS (6800 ± 2700 min^−1^ M^−1^), they are similar in magnitude and lie within error of each other.

The rate at which the absorbance of *Sa*SufS at 340 nm increased in the presence of different concentrations of Cys was also used to determine the relative *k_cat_* (*k_int_*) and *K_m_* values paralleling the formation of observable intermediates of the *Sa*SufS cysteine desulfurase catalytic cycle. The intervals of 0 to 40 s and 40 to 120 s were chosen to represent the short- and long-term rates of intermediate formation, as both intervals offered a reasonable range for precision. Additionally, preliminary data suggested a noticeable difference in the rate of change in the absorbance at 340 nm after 40 s in the absence and presence of TCEP. The *k_int_* values corresponding to the initial rate of intermediate formation (0 to 40 s) in the absence (350 ± 18 min^−1^) and presence (350 ± 13 min^−1^) of TCEP were identical. These results suggest that the initial formation of the absorbing intermediate does not require the reduction of Cys389 and, therefore, corresponds to a process occurring almost exclusively during the first turnover of the enzyme. Conversely, the data corresponding to the long-term rate of intermediate formation (40 to 120 s) in the absence of TCEP yielded a *k_int_* value (205 ± 8 min^−1^), which is larger than that observed in the presence of TCEP (173 ± 8 min^−1^). These results indicate the continued accumulation of a trapped intermediate upon the second turnover of the enzyme only if the reduction of Cys389 does not occur.

The *K_m_* data derived from the same Michaelis–Menten curves further corroborate these conclusions. The *K_m_* values corresponding to the initial rate of intermediate formation in the absence and presence of TCEP (570 ± 100 and 337 ± 52 μM, respectively), which likely represent the natural substrate affinity for *Sa*SufS, agree well with both each other and the *K_m_* values determined for the entire catalytic cycle of *Sa*SufS and *Bs*SufS using the Ala-NDA assay (600 ± 170 µM and 440 ± 190 μM, respectively). Conversely, the *K_m_* values corresponding to the long-term rate of intermediate formation in the absence and presence of TCEP (1030 ± 120 and 616 ± 93 μM, respectively) significantly differed from each other. Since TCEP is able to reduce the persulfide formed on Cys389 after the first turnover (and subsequent turnovers) of the enzyme, the latter *K_m_* likely represents the steady-state substrate affinity for *Sa*SufS. Indeed, it also agrees well with the *K_m_* for Cys reported for *Sa*SufS using the Ala-NDA assay. The former *K_m_* likely corresponds to the substrate affinity of persulfurated *Sa*SufS, which has undergone one complete turnover but was not able to be reduced and was subsequently trapped during its second turnover. These data suggest that the presence of persulfide in the active site decreases the binding affinity of Cys to *Sa*SufS. In addition to providing valuable kinetic information regarding the development of well-studied intermediates of the cysteine desulfurase catalytic mechanism, such observations ultimately demonstrate the capacity of this spectrophotometric kinetic assay (observation of the absorbance at 340 nm) to be used in measuring the cysteine desulfurase activity of *Sa*SufS.

### 3.2. Spectrophotometric Quantitation of the Efficacy of PLP-Binding Inhibitors of SaSufS

While the selectivity of aminotransferases is directed by the complex tertiary environments of their active sites, their ability to catalyze specific transamination reactions is ultimately mediated by the nonselective electrophilic Schiff base present within the structure of the internal Lys–aldimine (LLP) [42]. As such, primary amines with some affinity for the active site of an aminotransferase (e.g., DCS, LCS, β-chloroalanine, and myriocin) may, when introduced to that aminotransferase, react with LLP, forming a transient external aldimine that competes with the natural amino acid-substrate of the enzyme [30,43]. Upon formation of an external aldimine with LLP, some of these molecules, including DCS and LCS, may further interact with proximal active site residues, thermodynamically trapping the enzyme in a stable, nonreactive state [27,28,30]. In this manner, DCS and LCS uniquely behave as both competitive and suicide inhibitors of PLP-dependent aminotransferases [30].

The spectra of *Bs*SufS incubated with LCS, reported by Nakamura et al. [27], revealed a gradual decay of the LLP-associated absorbance at ~425 nm, concomitant with the emergence of a transient peak at ~380 nm. This absorbance was attributed to the external LCS–aldimine complex [27]. Over the course of several hours, the transient absorbance at ~380 nm gradually decreased in intensity, with the concomitant emergence of an absorption band at ~335 nm assigned to PMP–isoxazole adduct [27]. The absorbance data presented in Figure 4a indicate that, as previously observed in *Bs*SufS [27], LCS reacts with LLP to form an external LCS–aldimine (λ_max_ ~390 nm), which is then gradually converted to the stable aromatic PMP–isoxazole (λ_max_ ~335 nm) through an LCS–ketimine intermediate (Figure 2). The observed shift in λ_max_ of the LCS–aldimine-associated absorbance can be attributed to the decrease in absorbance of the relatively broad PLP-associated peak at 420 nm, which red-shifts the LCS–aldimine-associated peak, and the simultaneous emergence of the PMP-isoxazole-associated absorbance at ~335 nm. While both the formation of LCS–aldimine and the conversion of aldimine to PMP–isoxazole were markedly slower in *Sa*SufS than in *Bs*SufS [27], the emergence of an isoxazole-associated peak after only ~30 min, as well as the lack of an isosbestic point between the LLP and LCS–aldimine-associated peaks, suggests that the formation of aldimine, and therefore, binding of LCS, was significantly slower in *Sa*SufS.

The corresponding spectra, taken upon incubation of *Sa*SufS in DCS, revealed a single spectrophotometrically observable process resulting in a new peak at λ_max_ ~380 nm [27]. In *Bs*SufS, Nakamura et al. [27] saw a similar absorption emerge at ~385 nm, coupled with a drop in the PLP-associated ~425 nm peak. They ascribed the formation of this peak to the accumulation of an external DCS–aldimine complex [27]. However, they also observed a gradual blue-shift in the λ_max_ of the ~385 nm absorbance to ~360 nm over the course of 24 h, a feature that was attributed to the conversion of the DCS–aldimine intermediate to PMP and β-aminooxyacetaldehyde [27]. It is possible that, as was the case with LCS, DCS reacts significantly slower with *Sa*SufS than it does with *Bs*SufS since a significant shift in the aldimine-associated peak was observed only after the PLP-associated peak at ~425 nm had substantially decayed [27].

Due to their slow binding, varying concentrations of LCS and DCS were incubated with *Sa*SufS for 96 h before cysteine was added and the change in absorbance at 340 nm was observed as a function of time. Dose–response curves generated from these data revealed vastly different IC_50_ values for each enantiomer. The IC_50_ of LCS (62 ± 23 μM) was significantly lower than that of DCS (2170 ± 920 μM), indicating that LCS was a more effective inhibitor of *Sa*SufS. This observation was not unexpected, as LCS stereochemically mimics the natural Cys substrate of *Sa*SufS. Interestingly, Charan et al. [28] reported an IC_50_ value of 29 μM for the DCS-mediated inhibition of *P. falciparum* SufS, which was much lower than that obtained for *Sa*SufS. While the activity data at higher concentrations of the LCS dose–response curve could not be obtained using the Ala-NDA assay due to the reactivity between NDA and LCS, the activity data obtained at lower concentrations (IC_50_ = 33 ± 12 μM) agreed remarkably well with those obtained spectrophotometrically (Figure 5a). These results further demonstrate the accuracy and utility of the spectrophotometric assay. Based on these data, the spectrophotometric assay described herein is able to extract IC_50_ values in a simple and relatively fast manner, providing information regarding potential inhibitors of SufS enzymes. Furthermore, the discovery that LCS and DCS inhibit *Sa*SufS provides evidence that inhibitors targeting the PLP cofactor of cysteine desulfurase enzymes requisite to the biosynthesis of Fe-S clusters are viable targets for further therapeutic antibiotic development [44].

Interestingly, despite finding that both LCS and DCS irreversibly bind the PLP cofactor of *Bs*SufS, Nakamura et al. [27] demonstrated, via mass spectrometry and X-ray crystallography, that the incubation of *Bs*SufS with LCS and DCS yielded unique, nonreactive states. For the LCS-mediated inhibition of *Bs*SufS, the external aldimine was proposed to proceed to the corresponding ketimine through a quinonoid intermediate, as per the currently accepted SufS cysteine desulfurase reaction mechanism (Figure 1) [27]. The isoxazolidine C5 of the LCS-PLP external ketamine is then deprotonated by a nearby basic lysine residue, aromatizing the isoxazolidine ring and forming the observed stable pyridoxamine-5′-phosphate (PMP)–isoxazole adduct (Figure 2a) [27]. While the reversibility of the formation of this adduct has recently been reported in the alanine racemase from *M. tuberculosis*, a type III fold aminotransferase [45], crystallographic data have demonstrated that PMP–isoxazole often represents a thermodynamic sink from which SufS and other aminotransferases are unable to return [27,46,47,48]. For DCS, based on the mechanism of the inhibition of serine palmitoyltransferase [49], Nakamura et al. proposed that the external DCS–aldimine undergoes a ring-opening reaction, then a decarboxylation, ultimately forming β-aminooxyacetaldehyde and the observed product, PMP [27]. However, based on the proposed mechanism of the inhibition of the human aminotransferase, alanine-glyoxalate transaminase, it is likely that the ring opening of DCS occurs without a subsequent decarboxylation, ultimately yielding PMP and O-aminopyruvate (Figure 2b) [50].

### 3.3. S. aureus Growth Studies

*S. aureus* growth studies in the presence of LCS and DCS revealed that each was able to impede the growth of *S.* aureus with in vivo IC_50_ values of 4.4 ± 0.7 mM and 120 ± 13 μM for LCS and DCS, respectively. The IC_50_ of DCS is likely lower than that of LCS because DCS also effectively inhibits D-aminotransferases, D-alanine ligase and D-alanine racemase, both of which are necessary for peptidoglycan synthesis [51]. Indeed, the growth of the Δ*nfu* Δ*sufT* strain, which would be expected to exhibit greater overall sensitivity to the inhibition of Fe-S cluster biogenesis, revealed a weak inhibition by LCS, relative to the WT strain. As such, the development of modified inhibitors with a stronger affinity for the active site of SufS represents the next step toward the development of antibiotics targeting the Fe-S biogenesis pathway of *S. aureus*. These data demonstrate that the development of such antibiotics targeting the essential sulfur mobilization pathways of not just *S. aureus*, but also other SUF and SUF-like system-dependent pathogenic organisms, is a viable endeavor.

## 4. Materials and Methods

### 4.1. Materials

All of the chemicals were purchased from commercial sources of the highest quality available. The *Sa*SufS plasmid was developed by the Boyd lab at Rutgers University (New Brunswick, NJ, USA). BL-21(DE) cells, kanamycin, Luria–Bertani (LB) Broth, imidazole, nitrocellulose syringe filters, and tris(hydroxymethyl)aminomethane hydrochloride (Tris-HCl) were purchased from Thermo Fisher Scientific (Waltham, MA, USA). Isopropyl-β-D-1-thiogalactopyranoside (IPTG), potassium chloride (KCl), L-cysteine, L-alanine, trichloroacetic acid (TCA), boric acid, sodium hydroxide, potassium cyanide, hydrochloric acid, L-cycloserine, D-cycloserine, and glycerol were purchased from Sigma Aldrich (St. Louis, MO, USA). Pyridoxal 5′-phosphate (PLP), 3-(*N*-Morpholino)propanesulfonic acid (MOPS), tris(2-carboxyethyl)phosphine (TCEP), and 2,3-napthalenedicarboxyaldehyde (NDA) were purchased from Ambeed (Arlington Heights, IL, USA). The 5 mL IMAC Ni-NTA purification columns were purchased from Cytiva (Marlborough, MA, USA).

### 4.2. Plasmids, Cell Cultures, and Protein Purification

The *Sa*SufS plasmid was transformed into the BL-21(DE) *E. coli* strain in the presence of kanamycin at a final concentration of 50 μg mL^−1^ [20]. A 100 mL starter culture containing LB broth, 50 μg mL^−1^ of kanamycin, and the *Sa*SufS expression system was grown overnight at 37 °C. The next day, the culture was diluted in 800 mL with LB broth containing 50 μg/mL of kanamycin and grown at 37 °C until reaching an OD_600_ of 0.8–1. The cells were supplemented with PLP to a final concentration of 625 μM. The cells were then induced with a final concentration of 0.2 mM IPTG. After PLP supplementation and induction, the cells were grown for 20 h at 20 °C.

The cells were harvested by centrifugation at 13,974× *g* for 15 min using an Avanti J-26 XP centrifuge (Beckman Coulter, Inc., Brea, CA, USA). The cells were lysed in buffer A (50 mM Tris-HCl, pH 7.8, and 500 mM KCl with 20 mM imidazole) and centrifuged at 29,100× *g* at 4 °C for 55 min. The supernatant was filtered through a 0.45 μm nitrocellulose syringe filter before loading onto a HisTrap HP column (Cytiva). After washing with five column volumes of buffer A, the protein was eluted with a linear gradient of 0–100% of buffer B (50 mM Tris-HCl, pH 7.8, and 500 mM KCl, with 250 mM imidazole). The fractions were collected, concentrated, and buffer exchanged into buffer C (100 mM MOPS, pH 8.0, with 10% glycerol) using an Amicon Ultra-15 Centrifugal filter (30 kDa) (Millipore, Darmstadt, Germany) and an Eppendorf 5810 R centrifuge (Eppendorf, Hamburg, Germany) by spinning at 3200× *g* until the sample volume was less than 1 mL. The buffer exchange process was repeated three times. An SDS-polyacrylamide gel electrophoresis (SDS-PAGE) revealed a single polypeptide band at ~48 kDa, which was consistent with that in *Sa*SufS. The purified protein was aliquoted, flash-frozen, and stored at −80 °C.

### 4.3. Quantitation of SaSufS PLP Occupancy

The PLP occupancy was determined as described previously, with minor modifications [20]. Varying amounts of *Sa*SufS were diluted in 100 mM MOPS, pH 8.0, with 10% glycerol to 800 μL. A total of 200 μL of 5 M NaOH was added, and the samples were incubated at 75 °C for 10 min. Then, 85 μL of 12 M HCl was added, and the samples were centrifuged at 21,130× *g* for 5 min using an Eppendorf 5424 centrifuge (Eppendorf, Hamburg, Germany). The supernatant was transferred to a cuvette and the absorbance was measured at 390 nm. A standard curve of known PLP concentrations ranging from 0 to 100 μM under identical experimental conditions was used to quantify the PLP concentrations.

### 4.4. Ultraviolet–Visible Absorbance Spectroscopy

All spectra and time course UV-Vis data were acquired using a UV-2600i UV-Vis spectrophotometer (Shimadzu, Kyoto, Japan). Semi-micro quartz cuvettes were purchased from Starna Cells Inc. (Atascadero, CA, USA). Prior to each experiment, *Sa*SufS was thawed on ice and centrifuged at 18,312× *g* for 5 min in an Eppendorf 5424 desktop centrifuge to remove any precipitated protein. The protein was then transferred to a clean Eppendorf tube and thoroughly mixed. The concentration of the *Sa*SufS stock was subsequently determined by measuring the absorbance at 280 nm using a calculated molar extinction coefficient of 41,830 cm^−1^M^−1^ [52]. The quartz cuvettes were washed once with 18 MΩ water, once with methanol, then twice with 18 MΩ water before drying with filtered compressed air.

### 4.5. Cysteine Desulfurase Substrate and Inhibitor Incubation

All of the cysteine desulfurase incubation studies were performed with 4.9 mM Cys or inhibitor and 80–120 μM *Sa*SufS in buffer C. Prior to collecting the first spectrum in each experiment, the baseline absorbance was corrected to zero using a 1 mL sample and reference of 5 mM Cys or inhibitor in buffer C. A total of 20 μL of *Sa*SufS was mixed into 980 μL of 5 mM Cys and inhibitor in buffer C, and the UV-Vis spectrum was taken immediately after mixing. The UV-Vis spectra of *Sa*SufS in the presence of Cys were acquired once every 20 s for 5 min, while the UV-Vis spectra of *Sa*SufS in the presence of LCS and DCS were acquired once every hour for 24 h. The LCS spectra were additionally acquired once every 5 min for 60 min. In each case, a spectrum of *Sa*SufS alone was also acquired by mixing 20 μL of *Sa*SufS from the same stock solution with 980 μL of buffer C.

### 4.6. Cysteine Desulfurase Rate Dependence on Substrate Concentration

The cysteine desulfurase activity of *Sa*SufS was spectrophotometrically assessed as a function of Cys concentration. Nineteen 1 mL Cys solutions ranging in concentration from 0 to 60 mM were prepared in 1.5 mL Eppendorf tubes from a common 100 mM stock solution. For each Cys solution, 400 μL of *Sa*SufS diluted to 93.75 μM in 100 mM MOPS, pH 8.0 were mixed. Next, 100 μL of the respective Cys solutions were then mixed with *Sa*SufS, providing a final concentration of *Sa*SufS of 75 μM. Immediately after mixing, the absorbance at 340 nm was recorded for 2 min. The slopes were plotted as a function of Cys concentration from 0 to 120 s. All of the reactions were performed at 12 °C, as set by a TC 1 Temperature Controller (Quantum Northwest, Liberty Lake, WA, USA). All of the *Sa*SufS solutions were allowed to equilibrate at this temperature in the spectrometer before the reactions were initiated. All of the Cys solutions were prepared on the same day that the data were collected. The Cys solutions prepared for the trials involving 2 mM TCEP were made in buffer comprising 100 mM MOPS, pH 8.0, and 10 mM TCEP. All of the data were taken in triplicate, and the *Sa*SufS activities for each trial were scaled according to PLP occupancy.

### 4.7. Quantitation of Cysteine Desulfurase Inhibitor Efficacy

The efficacy of each inhibitor was quantified via a UV-Vis time course using 500 μL solutions of each inhibitor in buffer C by varying the concentrations from 0 to 30 μM. A 100 μM *Sa*SufS stock solution was diluted in buffer C to 50 μM and incubated at 20 °C with shaking in a Gyromax 747R Incubator Shaker (Amerex Instruments, Inc., Concord, CA, USA) for 96 h, with each inhibitor at various concentrations. To spectroscopically measure the *Sa*SufS-inhibitor solutions post-incubation, 100 μL of 50 mM Cys was added to 400 μL of each *Sa*SufS-inhibitor solution. These solutions were rapidly mixed for 5 s before initiating a time-course collection by monitoring the absorbance at 340 nm. The slope of each line from 0 to 120 s was recorded, and the slopes derived from each solution were plotted against the log[inhibitor]. All of the inhibitor efficacy time-course data were collected at 25 °C and measured in triplicate.

### 4.8. Alanine Detection Assay

The cysteine desulfurase activity was determined using a 96-well plate, as previously reported [25,33,34]. Ala production was quantified using a developing reaction with NDA, which forms a fluorescent adduct with Ala. All of the assays were prepared using an Xplorer multichannel pipette (Eppendorf, Hamburg, Germany), a black 96-well plate, and plastic multichannel reservoirs. For the cysteine saturation curves, 120 mL reactions were carried out containing 100 mM MOPS at pH 8.0, 2 mM TCEP, and 50 μM *Sa*SufS or *Bs*SufS with varying concentrations of Cys (0–1500 mM). A total of 60 μL of each Cys concentration examined was added to 60 μL of 100 mM *Sa*SufS or *Bs*SufS to initiate the reaction, and 24 μL of 6% trichloroacetic acid (TCA) was added to quench the reactions at the 1 and 2 min timepoints (for a final concentration of 1% TCA). TCA was added to *Sa*SufS or *Bs*SufS prior to adding cysteine for the 0 min timepoint. For the cycloserine inhibition assays, prior to setting up the cysteine desulfurase assay reaction, 50 μM of *Sa*SufS was incubated with varying amounts of inhibitor (0–30 mM LCS) for 96 h at 20 °C with shaking using an Excella E24 Incubator Shaker (New Brunswick Scientific Co., Inc., Edison, NJ, USA) or a Gyromax 747R Incubator Shaker (Amerex Instruments, Inc.). After 96 h of incubation, the *Sa*SufS and inhibitor samples were reacted with Cys. These reactions were carried out at room temperature, containing 100 mM MOPS at pH 8.0, 2 mM TCEP, 25 μM *Sa*SufS, and 1500 μM Cys. In total, 120 μL of the *Sa*SufS and inhibitor sample was combined with 120 μL of 3 mM Cys and 4 mM TCEP in a fresh Eppendorf tube to initiate the reaction. The reactions were quenched after 2 min by adding 48 μL of 6% TCA to finally obtain 1% TCA. The reaction solutions were then centrifuged at 21,130× *g* for 5 min using an Eppendorf 5424 centrifuge. A total of 144 μL of the supernatant of each sample were then transferred to the 96-well plate.

The quenched cysteine saturation curve samples and cycloserine inhibition samples were developed in an identical manner. A solution of 88 mM KCN and 880 mM sodium borate pH 9 was made immediately prior to setting up the developing reaction. A stock solution of NDA (dissolved in methanol) was combined with the KCN/sodium borate mix. A total of 56 μL of the NDA/KCN/sodium borate mix was added to each well to a final concentration of 2 mM NDA, 20 mM KCN, and 200 mM sodium borate. The plate was incubated in the dark for 30 min, and the resulting fluorescence was then measured using a Synergy Neo2 multimode plate reader (BioTek, Winooski, VT, USA) using an excitation wavelength of 390 nm and an emission wavelength of 440 nm. The Ala-NDA adduct was quantified using a standard curve of Ala under identical reaction conditions for each concentration of Cys or LCS to account for any background fluorescence from the unstable Cys-NDA adduct and the possible adduct formation between the primary amine of LCS and NDA.

### 4.9. Antimicrobial Assays

To determine the minimal inhibitory concentration (MIC) of the compounds, *Staphylococcus aureus* USA300_LAC and Δ*nfu* Δ*sufT* double mutant were grown overnight (~17 h) in 2 mL of Mueller–Hinton Broth (MHB) in 10 mL capacity culture tubes at 37 °C with agitation. The optical density at 600 nm of the overnight cultures was adjusted to 0.01 in MHB. In total, 100 μL of adjusted culture was subcultured into the wells of clear, polystyrene 96-well microtiter plates containing 100 μL of MHB with an inhibitor or vehicle control. LCS was prepared in 60 mM stock and DCS as a 12 mM stock. The compounds were serially diluted to give the final concentrations. The minimal inhibitory concentrations (MICs) and IC_50_ values were determined for three biological replicates. The control wells contained uninoculated MHB with an inhibitor or a vehicle control. The absorbance at 600 nm was measured after 18 h of inoculation using an Epoch 2 microplate reader (BioTek, Winooski, VT, USA).

## 5. Conclusions

This study utilized a previously reported Ala-NDA assay to validate a new spectrophotometric assay for *Sa*SufS, which entails the observance of absorbance at 340 nm for *Sa*SufS upon the addition of Cys to derive both the absolute and relative Michaelis–Menten kinetic constants. These kinetic constants were shown to correspond to intermediates within the cysteine desulfurase catalytic cycle in the presence of TCEP and the formation of what is likely the rate-limiting intermediate of the cysteine desulfurase reaction. Using these methods, the relative *k_cat_* and *K_m_* values for Cys were determined for persulfurated *Sa*SufS (in the absence of TCEP) and monosulfurated *Sa*SufS (in the presence of TCEP), respectively. The monosulfurated *K_m_* values agree well with those obtained using the Ala-NDA assay, while the *K_m_* value for the persulfurated intermediate has not yet been reported. The spectrophotometric feature at 340 nm upon the addition of Cys was further used to assess the efficacy of the PLP-binding covalent inhibitors DCS and LCS. These potential *Sa*SufS inhibitors exhibited dose–response curves resulting in IC_50_ values of ~2200 and ~50 μM, respectively. Finally, *S. aureus* growth studies demonstrated that both DCS and LCS are able to inhibit the growth of *S. aureus*, though the future modification of these inhibitors to target *Sa*SufS more selectively in vivo is needed.

## Data Availability

The datasets used and/or analyzed during the current study are available from the corresponding author upon reasonable request.

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
