# Peer review of "Development of a Spectrophotometric Assay for the Cysteine Desulfurase from Staphylococcus aureus"

_antibiotics, 2025, doi:10.3390/antibiotics14020129_

Round 1

Reviewer 1 Report

Comments and Suggestions for Authors

In this manuscript, Sabo and group described their development of a spectrophotometric assay which they used in characterizing Staphylococcal cysteine desulfarase (SufS). They stated that their method is superior to existing assays since the latter employs a chemical quenching procedure and a long incubation time, and has the possibility of side reactions that might impact the accuracy of the assays. Therefore, to circumvent these problems in profiling for SufS, the authors proposed the use of a more direct spectrophotometric assay that has a lower risk of side reactions. This assay is based on the change in absorbance of SufS when its natural substrate, cysteine, is available. Here, in the absence of cysteine, SufS has an absorption peak at 420 nm. This peak diminishes when cysteine is present, leading to the formation of a second peak at 340 nm. Using this assay, the authors calculated the Michaelis-Menten parameters for the PLP intermediate responsible for this absorbance change and compared it to what has been reported for other methods. Furthermore, they used this assay to determine the IC50 values for D-cycloserine and L-serine, the former being an antimycobacterial drug.

Comments and Questions:

1. Consider rewriting your title to make it simpler. This might help: “Development of a spectrophotometric method for the study of staphylococcal cysteine desulfurase”.

2. In the background/objectives section, the authors did not clearly state the problem. They should consider rewriting line 17 to say that “Current methods for measuring the activity of this protein have allowed for its characterization, but they are hampered by their use of chemical reagents that need a long incubation time and might cause unwanted side reactions. This problem highlights the need for the development of a rapid quantitative assay for the characterization of SufS”.

3. Make line 112 a more complete sentence by noting that it is the UV-Vis spectra of Staph SufS that are measured.

4. Line 113: Include the reference paper where you drew the conclusion that the spectra is typical of the PLP cofactor bound to Lys250.

5. Line 121 to 133, the authors should include the Km and Kcat values from the alanine-based method of the previous paper. They included this in Table 1, but they didn’t mention their past study in this section. Readers will appreciate a comparison of the values obtained from both the ala-NDA method and spectrophotometric assay.

6. In section 2.4, a 96-hour incubation was used in determining the IC50 values of DCS and LCS. This long period seems to defeat one of the stated aims of the spectrophotometric assay which is to develop an alternative assay that is quick. Can you reconcile this time period to the long incubation periods needed for the chemical methods? Additionally, the IC50 values of LCS and DCS seem high, and their binding kinetics seem long, possibly making them nonideal inhibitors for this assay. Are there other inhibitors of SufS that has been reported? What is the IC50 value for LCS that was determined with the ala-NDA method? The authors included the dose-response curves (for ala-NDA method) in Figure 4A but never provided the IC50.

7. The experiments described in section 2.5 is inconclusive in determining if LCS and DCS inhibit the growth of S. aureus through the inhibition of Suf-directed Fe-S cluster synthesis. This point is particularly important since the dose response curves of the inhibitors against the double mutant and WT looks the same, showing that the missing Fe-S proteins might not be playing a role in the growth inhibition. Can you reconcile this?

8. Consider including a docking analysis of SufS with LCS and DCS. 

Author Response

Comment 1. Consider rewriting your title to make it simpler. This might help: “Development of a spectrophotometric method for the study of staphylococcal cysteine desulfurase”.

        We agree with the reviewer and changed the title to “Development of a Spectrophotometric Assay for the Cysteine Desulfurase from Staphylococcus aureus.”

Comment 2. In the background/objectives section, the authors did not clearly state the problem. They should consider rewriting line 17 to say that “Current methods for measuring the activity of this protein have allowed for its characterization, but they are hampered by their use of chemical reagents that need a long incubation time and might cause unwanted side reactions. This problem highlights the need for the development of a rapid quantitative assay for the characterization of SufS”.

                We appreciate the reviewers comment and modified the sentence to reflect the reviewer’s suggestion: “Current methods for measuring the activity of this protein allow for its characterization, but they are hampered by their use of chemical reagents which require long incubation times and may cause undesired side reactions. This problem highlights a need for the development of a rapid quantitative assay for the characterization of SaSufS in the presence of potential inhibitors.”

Comment 3. Make line 112 a more complete sentence by noting that it is the UV-Vis spectra of Staph SufS that are measured.

                We thank the reviewer for their suggestion and changed the sentence to read: “UV-Vis spectra of SaSufS were recorded between 300 to 460 nm over the course of 5 min at 22 °C (Figure 1).”

Comment 4. Line 113: Include the reference paper where you drew the conclusion that the spectra is typical of the PLP cofactor bound to Lys250.

                We added references to three papers in which a similar peak at roughly 420 nm was ascribed to the internal Lys-aldimine of a related cysteine desulfurase as suggested by the reviewer.

Comment 5. Line 121 to 133, the authors should include the Km and kcat values from the alanine-based method of the previous paper. They included this in Table 1, but they didn’t mention their past study in this section. Readers will appreciate a comparison of the values obtained from both the ala-NDA method and spectrophotometric assay.

We thank the reviewer for their suggestion and have added these values in a separate results subsection, as well as within the discussion.

Comment 6. In section 2.4, a 96-hour incubation was used in determining the IC50 values of DCS and LCS. This long period seems to defeat one of the stated aims of the spectrophotometric assay which is to develop an alternative assay that is quick. Can you reconcile this time period to the long incubation periods needed for the chemical methods? Additionally, the IC50 values of LCS and DCS seem high, and their binding kinetics seem long, possibly making them nonideal inhibitors for this assay. Are there other inhibitors of SufS that has been reported? What is the IC50 value for LCS that was determined with the ala-NDA method? The authors included the dose-response curves (for ala-NDA method) in Figure 4A but never provided the IC50.

                We agree that the 96-hour incubation period used for DCS and LCS does not lend itself to a quick assay. The length of this incubation period, however, is solely a product of the slow binding speed of DCS and LCS, and as such, would be required for any cysteine desulfurase assay if the presented results were to be replicated. Faster binding inhibitors (i.e., better inhibitors of SaSufS) should require a much smaller incubation period. As such, we also agree that LCS and DCS are not ideal inhibitors of SaSufS but were used as “proof of concept” for verifying the assay. Verification of the IC50 value (33 ± 12 µM) of LCS was obtained using the Ala-NDA data and is presented in Figure 4A.

Comment 7. The experiments described in section 2.5 is inconclusive in determining if LCS and DCS inhibit the growth of S. aureus through the inhibition of Suf-directed Fe-S cluster synthesis. This point is particularly important since the dose response curves of the inhibitors against the double mutant and WT looks the same, showing that the missing Fe-S proteins might not be playing a role in the growth inhibition. Can you reconcile this?

We agree with the reviewer. We are currently unsure if DCS or LCS is inhibiting SufS in vivo. These data do demonstrate some inhibition of S. aureus growth by LCS in the strain that is defective in maturating Fe-S proteins. As DCS is used as an antibiotic to treat specific bacterial infections because it can inhibit some PLP-dependent enzymes—specifically D-alanine racemase, which functions in cell wall synthesis, DCS was not active enough against S. aureus to be an effective treatment. It is likely that DCS and LCS have more than one target in S. aureus cells, and our data suggest that SufS could be one of those targets for LCS; however, these data are not confirmatory, and we have been careful in the text not to over-interpret this observation.

Comment 8. Consider including a docking analysis of SufS with LCS and DCS. 

                We thank the reviewer for their suggestion. While we agree that such an analysis could be valuable, it is beyond the scope of the current manuscript.

Reviewer 2 Report

Comments and Suggestions for Authors

Sabo et al. described a novel spectrophotometric assay that circumvent the challenges posed by conventional Ala-NDA (nonspecific reactivity with primary amines). The assay has been validated to be robust for protein kinetics and compound inhibition measurement. The manuscript is recommended for publication with the following minor revisions:

1. In result 2.2, the authors reported SaSufS kinetics of short- and long-term that seems quite close in time (40 s and 120 s). Is the difference significant enough to distinguish between initial rate and long-term rate of intermediate formation?

2. In result 2.2, the authors summarized the kinetics profile of different time points and with/without TCEP compared to previously reported data. Could the authors also determine the kcat values as all 4 curves seems to reach plateau, so a kcat should be able to be extracted by comparing to standard curve. The comparison will establish this novel assay more robust.

3. In result 2.5, could the author rationalize why the antimicrobial activity of LCS and DCS demonstrated a different pattern than in vitro inhibition?

Author Response

Comment 1. In result 2.2, the authors reported SaSufS kinetics of short- and long-term that seems quite close in time (40 s and 120 s). Is the difference significant enough to distinguish between initial rate and long-term rate of intermediate formation?

                We thank reviewer for this question. While the short-term rates of intermediate formation with and without TCEP are similar, the data show that the long-term rate of formation without TCEP (40-120 seconds after the reaction was initiated) is significantly greater than that with TCEP, indicating the accumulation of an intermediate only when SufS is unable to be reduced. This is clearly stated in the manuscript. We have cited the literature where others have reported similar timescales.

Comment 2. In result 2.2, the authors summarized the kinetics profile of different time points and with/without TCEP compared to previously reported data. Could the authors also determine the kcat values as all 4 curves seems to reach plateau, so a kcat should be able to be extracted by comparing to standard curve. The comparison will establish this novel assay more robust.

We agree that determination of absolute kcat values through a molar extinction coefficient determined via a standard curve would be useful. Unfortunately, we cannot currently derive an accurate molar extinction coefficient because multiple transient intermediates are present. 

Comment 3. In result 2.5, could the author rationalize why the antimicrobial activity of LCS and DCS demonstrated a different pattern than in vitro inhibition?

                See comment 7 of reviewer 1.

Reviewer 3 Report

Comments and Suggestions for Authors

A rapid method of quantitative analysis of antibiotics by using a spectrophotometric assay to measure the activity of PLP-dependent cysteine de-sulfurase of the SUF-like iron sulfur cluster biogenesis pathway in the absence and presence of the PLP-binding inhibitors D-cycloserine and L-cycloserine was developed. The IC50 values for D-cycloserine and L-cycloserine were measured and yield the IC50 values for PLP-dependent cysteine de-sulfurase of the SUF-like iron sulfur cluster biogenesis pathway as 2,170 ± 920 and 62 ± 23 μM respectively.

This method is useful for finding the new antibiotics , and this manuscript can be accepted for publication.

Author Response

No comments.

                We thank the reviewer for recommending our manuscript be accepted as submitted.

Reviewer 4 Report

Comments and Suggestions for Authors

The authors have presented a well-researched study with significant contribution to searching for potent antimicrobial agent. However, I have the following comments for them to address before recommending acceptance.

i. Some of the abbreviations in the abstract are confusing. Authors should state the full meaning before using abbreviations

ii. In the introduction section, authors should provide data for 2023 or 2024. This will educate readers on the urgency of the study

iii. The last paragraph of the introduction section ought to emphasize the novelty of the study as well as the aim and objectives of the study not including the potential results expected as it has been presented in the last four lines of the paragraph.

iv. In the discussion section, authors should provide justification for the methods used in their study.

v. MIC study is not enough to justify the effect of the intervention agent used. Authors should perform time-kill assay to strengthen their work.

Comments on the Quality of English Language

The quality of English language can be improved

Author Response

Comment 1. Some of the abbreviations in the abstract are confusing. Authors should state the full meaning before using abbreviations

                We have updated the abstract such that all abbreviations are defined in full.

Comment 2. In the introduction section, authors should provide data for 2023 or 2024. This will educate readers on the urgency of the study

We agree with the reviewer that more up-to-date data would provide a better picture of the ongoing antibiotic crisis as it stands today. Unfortunately, the latest estimate of global deaths due to antimicrobial-resistant pathogens that we could find (published in 2024) covers only through 2021. We have included these data in the introduction. We have also opted to present the originally-included 2019 data, as they were gathered pre-COVID-19 and thus may better represent the severity of the ongoing antibiotic crisis during non-pandemic years. We have also added the projected resistant numbers in 2050.

Comment 3. The last paragraph of the introduction section ought to emphasize the novelty of the study as well as the aim and objectives of the study not including the potential results expected as it has been presented in the last four lines of the paragraph.

                We have altered the last paragraph of the introduction to more effectively convey the objectives of the study. We believe that the potential results in this section highlight the importance and novelty of the work. As such, we have opted to retain them.

Comment 4. In the discussion section, authors should provide justification for the methods used in their study.

                We have added a few sentences to the discussion to better justify the methods.

Comment 5. MIC study is not enough to justify the effect of the intervention agent used. Authors should perform time-kill assay to strengthen their work.

We thank the reviewer for their suggestion. While a time-kill assay would represent a great step toward testing the efficacy of new antibiotics targeting S. aureus, we believe that the current MIC study serves to adequately verify the inhibitory effects of these compounds. Both DCS and LCS are known inhibitors of PLP-dependent enzymes, and DCS is already used as an antibiotic. Our intentions in presenting these data are not to demonstrate that DCS and LCS may be used to treat S. aureus, but to confirm that DCS and LCS inhibit its growth likely via covalently binding essential PLP-dependent enzymes such as SaSufS and to provide “proof of concept” for our new assay.

Round 2

Reviewer 1 Report

Comments and Suggestions for Authors

Title: Change "from" to "of"

Line 12: Antibiotic-resistant

Line 14-16: The sentence looks incomplete. "...exclusively relies [for what?]"

Line 31: The sentence doesn't look readable. Make the sentence clearer as follows: "The spectrophotometric method was then utilized to determine the half-maximal inhibitory concentrations (IC50) of DCS and LCS against SaSufS as [value1] and [value1], respectively".

Line 223-230, line 426-428: This is in response to my original comment 7. The authors agreed with my comment and noted that they are not sure if DCS and LCS is inhibiting S. aureus growth through inhibition of SuFS. But they still went ahead in line 428 to note that DCS and LCS showed weak inhibition of the double mutant relative to the WT. This is not really clear in Figure 6. If the authors insist on including this, they should provide the EC50 values of DCS and LCS for the double mutant. They did this for the WT, but not for the double mutant (see line 229). 

Author Response

Reviewer 1:

Comment 1. Title: Change "from" to "of"

              We have left “from” as the enzyme is purified from S. aureus and not “of” it.

Comment 2. Line 12: Antibiotic-resistant

              We have added the hyphen as suggested.

Comment 3. Line 14-16: The sentence looks incomplete. "...exclusively relies [for what?]"

              The sentence refers to the fact that S. aureus relies exclusively on the SUF-Like pathway for Fe-S cluster synthesis.  We have edited it to be “of the SUF-like iron sulfur (Fe-S) cluster biogenesis pathway upon which S. aureus relies exclusively for Fe-S synthesis.”

Comment 4. Line 31: The sentence doesn't look readable. Make the sentence clearer as follows: "The spectrophotometric method was then utilized to determine the half-maximal inhibitory concentrations (IC50) of DCS and LCS against SaSufS as [value1] and [value1], respectively".

We have clarified the sentence to read “The spectrophotometric method was then utilized to determine half maximal inhibitory concentration (IC50) values for DCS and LCS binding to SaSufS, which are 2,170  920 and 62  23 μM, respectively.”

Comment 5. Line 223-230, line 426-428: This is in response to my original comment 7. The authors agreed with my comment and noted that they are not sure if DCS and LCS is inhibiting S. aureus growth through inhibition of SuFS. But they still went ahead in line 428 to note that DCS and LCS showed weak inhibition of the double mutant relative to the WT. This is not really clear in Figure 6. If the authors insist on including this, they should provide the EC50 values of DCS and LCS for the double mutant. They did this for the WT, but not for the double mutant (see line 229). 

              We appreciate the reviewer’s comments but as is stated in the manuscript, it is weak inhibition and the two data sets in Figure 6a are different and not within error of each other. As such, we edited the manuscript in line 428 to only say that LCS is a weak inhibitor.

Reviewer 4 Report

Comments and Suggestions for Authors

No comment

Author Response

We thank the reviewer for accepting our manuscript.